# The Prevalence and Antibiotic-Resistant of *Listeria monocytogenes* in Livestock and Poultry Meat in China and the EU from 2001 to 2022: A Systematic Review and Meta-Analysis

**DOI:** 10.3390/foods12040769

**Published:** 2023-02-10

**Authors:** Haoqi Zhang, Xin Luo, Zafeiro Aspridou, Ourania Misiou, Pengcheng Dong, Yimin Zhang

**Affiliations:** 1Laboratory of Beef Processing and Quality Control, College of Food Science and Engineering, Shandong Agricultural University, Tai’an 271018, China; 2National R&D Center for Beef Processing Technology, Tai’an 271018, China; 3Laboratory of Food Microbiology and Hygiene, Department of Food Science and Technology, Faculty of Agriculture, Forestry and Natural Environment, School of Agriculture, Aristotle University of Thessaloniki, 541 24 Thessaloniki, Greece

**Keywords:** *Listeria monocytogenes*, livestock and poultry meat, prevalence, antibiotic resistance, meta-analysis

## Abstract

To compare the prevalence and antibiotic resistance rate of *Listeria monocytogenes* in livestock and poultry (beef, pork and chicken) meat between China and the European Union (EU), a meta-analysis was conducted. Ninety-one out of 2156 articles in Chinese and English published between January 2001 and February 2022 were selected from four databases. The prevalence of *L. monocytogenes* in livestock and poultry (beef, pork and chicken) meat in China and Europe was 7.1% (3152/56,511, 95% CI: 5.8–8.6%) and 8.3% (2264/889,309, 95% CI: 5.9–11.0%), respectively. Moreover, a decreasing trend was observed in both regions over time. Regarding antibiotic resistance, for the resistance to 15 antibiotics, the pooled prevalence was 5.8% (95% CI: 3.1–9.1%). In both regions, the highest prevalence was found in oxacillin, ceftriaxone and tetracycline, and a large difference was reported between China and the EU in ceftriaxone (52.6% vs. 17.3%) and cefotaxime (7.0% vs. 0.0%). Based on the above, it remains a significant challenge to enforce good control measures against the meat-sourced *L. monocytogenes* both in China and in the EU.

## 1. Introduction

*Listeria monocytogenes* is a bacterial pathogen that is mainly transmitted through food. The pathogen is notorious worldwide, and can grow in high salinity (10%), low water activity (<0.9), low temperature (4 °C), and in a pH range of 4.1–9.6 [1]. Therefore, it can survive in processing equipment, packaging materials, food contact surfaces, etc. Food processing, in particular, is considered to be the most important pathway for *L. monocytogenes* contamination [2]. *L. monocytogenes* can cause disease in certain groups, such as the elderly, infants, pregnant women, and people with weakened immunity, with a fatality rate of up to 30% in some countries [3,4], including the European Union (EU), the United States, and China [5,6,7].

At present, *L. monocytogenes* is frequently found in a variety of food products, especially those of animal origin, with a plethora of isolated strains being resistant to antibiotics [8,9]. Therefore, various foods or food products, especially meat products, have been investigated in recent years [10]. In 2018, the highest contamination rate of *L. monocytogenes* in the EU was reported for RTE (ready to eat) food products, accounting for 37.5% of all contamination [4]. From 2016 to 2018, *L. monocytogenes* was detected in 5996 raw meat products in Poland, with a prevalence rate of 2.1% and the rate among different types of meats varying [11]. The prevalence of *L. monocytogenes* in livestock and poultry meat in 28 provinces in China has been investigated, with the highest prevalence of 8.91% in meat and poultry [12]. The prevalence of *L. monocytogenes* in ruminant farms and slaughter environments in China was also tested, and *L. monocytogenes* was detected in multiple processes in the slaughter environment [13]. Hence, authorities around the world were devoted to the study of the prevalence of *L. monocytogenes* in meat products, covering different types of meat and over different periods, and even incorporating research studies in the production chain [14,15]. However, based on case studies, there are still differences in prevalence between countries and even between cities, and it is hard to see the whole picture in terms of the prevalence of this pathogen in a certain area.

Meanwhile, the Food and Agriculture Organization (FAO), the World Organization for Animal Health (WOAH)), and the World Health Organization (WHO) have collected and provided data about the emergence of antibiotic resistant *L. monocytogenes* isolates, and the hazard of *L monocytogenes* in food was further elaborated upon [16]. *L. monocytogenes* isolated in different countries developed resistance to first-line antibiotics such as rifampin, kanamycin, streptomycin and erythromycin [17,18,19,20]. The heavy use of prescription drugs in livestock and clinics is one of the reasons for the increased frequency of antibiotic resistance [10]. Additionally, multi-drug resistant strains were beginning to emerge in food and environmental isolates [21]. Some case studies found a high frequency of antibiotic resistance to cefotaxime, ciprofloxacin and tetracycline in certain *L. monocytogenes* [22,23,24,25,26]. Another study found that all 25 strains, isolated from pig slaughtering sites in Romania, were resistant to penicillin, imipenem and fusidiac acid and seven other antibiotics [27]. However, the overall understanding of the prevalence of the antibiotic resistance of *L. monocytogenes* in various geographical regions has also been less studied.

In recent years, more and more meta-analyses have been conducted in the field of food safety [28,29,30,31,32,33]. A meta-analysis can deal with the overall and sub-sets of the prevalence of certain pathogens in food and show the pre- and post-intervention effects on food microbes, and it is therefore a powerful tool for evaluating, identifying and summarizing the results of a large number of studies [34]. A meta-analysis of the prevalence of *Listeria* spp. in foods was performed in Iran, and a pooled prevalence of *L. monocytogenes* in meat products of 2.6% (95% CI: 0.2–35.0%) was reported [35]. In another study, a meta-analysis of the prevalence of *L. monocytogenes* in meat products in China was conducted from 2007 to 2017, and it found that the prevalence of *L. monocytogenes* in raw and RTE meat products was 8.5% (95% CI: 7.1–10.3%) and 3.2% (95% CI: 2.7–3.9%), respectively [34]. This is despite the fact that some meta-analyses of *L. monocytogenes* were already conducted in meat products and clear information on the prevalence of the pathogen in different types of meat was provided. China and the EU can partially reflect the meat safety levels for the developing and developed countries since both of them are big meat consumers and producers [36,37]. Considering that no meta-analysis has been conducted on the prevalence and antibiotic resistance of *L. monocytogenes* in livestock and poultry meat products between China and the EU over a 20-year period to date, an updated systematic comparative review on the prevalence and antibiotic resistance of *L. monocytogenes* in these regions is urgently needed.

Therefore, the objective of this meta-analysis was to assess the prevalence and antibiotic resistance of *L. monocytogenes* in livestock and poultry (beef, pork and chicken) meat products in China and the EU by extracting data from numerous literature sources collected from various databases. In the quantitative analysis, the differences between China and the EU were compared through the data and the discrepancies were discussed. The results obtained in this study can further help to formulate reasonable preventive measures against *L. monocytogenes* and to select effective drugs for the treatment of listeriosis through the understanding of antibiotic resistance.

## 2. Materials and Methods

### 2.1. Literature Collection

The Cochrane protocol was followed for this meta-analysis [38]. The PRISMA statement [39] was employed for reporting the screening process. In early 2022, four databases in English and Chinese were used for literature retrieval, including PubMed (accessed on 18 February 2020), Web of Science (accessed on 18 February 2020), CNKI (www.cnki.net, in Chinese) (accessed on 18 February 2020) and the Wanfang Data knowledge service platform (www.wanfangdata.com.cn, in Chinese) (accessed on 18 February 2020). The search terms used in the literature base are shown in Table 1. In the Chinese databases, a synonym extension was used for the search. In the English databases, MeSH (Medical Subject Headings) was used to obtain the subject terms and synonyms associated with the search terms. The publication time of the articles was set between 2001 and 2022.

### 2.2. Inclusion/Exclusion Criteria

In order to exclude or include an article, the following criteria were applied: (1) The sampling time should be clearly stated; (2) Sampling sites should be identified (China or EU); (3) Sample types including livestock and poultry (beef, pork and chicken) meat products were reported; (4) Sample size should be reported; (5) Samples should be tested for the prevalence or antibiotic resistance of *L. monocytogenes*; and (6) Chinese articles should be collected in the Chinese Science Citation Database (CSCD).

### 2.3. Data Extraction

The retrieved documents were imported into Endnote 20 software (Clarivate Inc., Philadelphia, PA, USA) and were kept after removing duplicates. The data in each selected study were extracted, including the following basic information: author name, publication time, sampling time, sampling location, sample name, total sample size, number of detections of *L. monocytogenes*, and antibiotic resistance information. The extract data were entered into a Microsoft Excel spreadsheet for data analysis and quality assessment. The basic information extracted from the article was compared with the criteria set for doing so, and those that met all of the criteria were included. If one of the above criteria was not met, the article was excluded. The data were manually extracted and evaluated by two authors, and the differences in opinion were referred to a third person for evaluation [40].

### 2.4. Meta-Analysis and Statistical Analysis

The total sample size, the number of samples contaminated with *L. monocytogenes*, the number of *L. monocytogenes* strains isolated from meat and meat products and those showing antibiotic resistance, as well as the prevalence and antibiotic resistance rate of *L. monocytogenes* were meta-analyzed. The original data were tested for normal distribution prior to the analysis. When the original data did not conform to the normal distribution, the data were transformed using a total of logarithmic transformation, logit transformation, inverse sine transformation, and Freeman-Tukey transformation and then tested again for normal distribution. According to the test results, the Freeman-Tukey transformed data was closest to the normal distribution.

All of the data were processed by Stata17 software (StataCorp LLC, College Station, Texas, USA). The included data were processed to obtain the overall prevalence and antibiotic resistance of *L. monocytogenes*. Heterogeneity was then assessed, mainly by using the I-square test and the associated significance *p*-value for quantitative assessment. For studies with high heterogeneity (*I*^2^ > 50%), a subgroup analysis was performed to determine the source of the heterogeneity. When *p*-value < 0.10 and *I*^2^ > 50%, a random effects model was used for meta-analysis; otherwise, a fixed effects model was used [41]. The 95% confidence interval of the combined effect size was then calculated. A funnel plot was then drawn to determine if there was any bias, and an Egger test was performed. In data processing, the Metan and the Metaprop were normally used [42,43]. Since there were cases where the prevalence or detection rate is zero or one in the data, the Metan in the software could not be used for processing because the cases of zero and one were excluded. Therefore, the Metaprop is used in combination with Freeman-Tukey conversion processing. The ability to include effect sizes of zero or one ensures the integrity and credibility of the results.

## 3. Results

### 3.1. Article Inclusion

The literature search and inclusion process are shown in Figure 1. A total of 2156 articles were retrieved from the four selected electronic databases. According to the established criteria, the articles were screened in detail, and finally 91 articles were determined to be in compliance with the established criteria (47 and 44 in Chinese and English, respectively). Although the included articles were published in 2001–2022, the sampling was performed between 2000 and 2020. The sampling sites covered 14 EU countries and 21 provinces, cities, and autonomous regions in China.

### 3.2. Pooled Prevalence of L. monocytogenes in Livestock and Poultry Meat in China and the EU

The pooled prevalence of *L. monocytogenes* in livestock and poultry (beef, pork and chicken) meat in China and the EU was 7.4% (95% CI: 5.9–9.6%, *I*^2^ = 98.72%, *t*^2^ = 0.054, *p* < 0.01), and the results showed high heterogeneity. A subgroup analysis was needed to analyze the sources of the heterogeneity. Regarding the assessment of risk of bias, the funnel plot in Figure 2 is not asymmetric, with some points falling outside of the funnel. The results of an Egger test illustrated in Figure 3 show that the regression intercept was far from the zero point. This indicates that there is a high degree of publication bias.

#### 3.2.1. Pooled Prevalence of *L. monocytogenes* in Each Region

The included publications were divided into subgroups by region (Table 2). The pooled prevalence in China was 7.1% (95% CI: 5.8–8.6%), which was lower than the 8.3% (95% CI: 5.9–11.0%) of the EU (*p* < 0.01).

#### 3.2.2. Pooled Prevalence of *L. monocytogenes* in Different Times

In order to determine whether the prevalence of *L. monocytogenes* was changing over time, subgroups were created based on sampling times in 2000–2005, 2006–2010, 2011–2016, and 2017–2020. As shown in Table 3, the pooled prevalence of *L. monocytogenes* in livestock and poultry (beef, pork and chicken) meat generally showed a downward trend, with the prevalence in 2000–2005, 2006–2010, 2011–2016 and 2017–2020 being 9.1% (95% CI: 6.0–12.9%), 7.4% (95% CI: 5.6–9.8%), 6.6% (95% CI: 4.6–8.8%), and 6.7% (95% CI: 3.1–11.5%), respectively. Although a slight upward trend in prevalence was observed from 2017 to 2020, the trend needs to be verified over time given the limited number of studies for this period.

#### 3.2.3. Pooled Prevalence of *L. monocytogenes* between Raw Meat and RTE or Cooked Meat

Due to the wide variety of food products in each territory, it was not possible to carry out a detailed classification of meat products. Therefore, only the products that were clearly identified in the original articles were included for the following subgroup analysis, including raw meat and RTE or cooked meat. As presented in Table 4, the pooled prevalence of *L. monocytogenes* in raw meat was 11.3% (95% CI: 9.0–13.8%) much higher than that in RTE or cooked meat, with a prevalence of 3.7% (95% CI: 2.9–4.7%) (*p* < 0.01). It was also found that the prevalence in raw meat in China (10.1%, 95% CI: 7.8–12.7%) was lower than that in the EU (16.8%, 95% CI: 9.9–24.9%) (*p* < 0.01). The prevalence in RTE or cooked meat in China (3.5%, 95% CI: 2.6–4.5%) was slightly lower than that in the EU (4.1%, 95% CI: 2.5–6.0%) (*p* < 0.01).

#### 3.2.4. Pooled Prevalence of *L. monocytogenes* among Pork, Beef and Chicken Meat

In Table 5, among the publications included, the highest prevalence of *L. monocytogenes* was found in chicken and pork, with 10.5% (95% CI: 6.8–14.9%) and 10.5% (95% CI: 6.4–15.3%), respectively, while the lowest prevalence was 8.6% (95% CI: 5.4–12.3%) in beef (*p* < 0.01). Regarding the prevalence of *L. monocytogenes* in various meat products in China and the EU, the prevalence in pork in China was higher than that in the EU (11.1%, 95% CI: 6.6–16.4% vs. 8.8% 95% CI: 2.1–19.2), but the opposite was observed for chicken (China: 9.9%, 95% CI: 5.8–14.8% vs. EU: 13.9%, 95% CI: 1.3–35.6%), while the prevalence in beef was at the same level (China: 8.7%, 95% CI: 4.8–13.4% vs. EU: 9.0%, 95% CI: 3.5–16.6%) in both regions (*p* < 0.01).

#### 3.2.5. Pooled Prevalence of *L. monocytogenes* by Different Detection Methods

The detection method could be regarded as a source of deviation in the reported prevalence of *L. monocytogenes*. According to Table 6, when biochemical identification was used as the detection method, the prevalence was 7.0% (95% CI: 5.7–8.4%), while the prevalence reached 9.0% (95% CI: 5.7–13.0%) when molecular detection methods were used.

### 3.3. Pooled Prevalence of Antibiotic Resistance of L. monocytogenes

A total of nine articles about antibiotic resistant *L. monocytogenes* were included in the meta-analysis. The pooled prevalence of the antibiotic resistance of *L. monocytogenes* in livestock and poultry (beef, pork and chicken) meat was 5.8% (95% CI: 3.1–9.1%), including China and the EU. The heterogeneity in the studies was 89.87%. According to the funnel plot (Figure 4), most of the points inside were (relatively) symmetrically distributed and a *p*-value equal to 0.249 (*p* > 0.05) was obtained by the Egger test (Figure 5). Thus, it can be concluded that the nine studies included have no or limited publication bias.

#### 3.3.1. Pooled Prevalence of Antibiotic Resistance of *L. monocytogenes* in Livestock in Different Regions

The prevalence of antibiotic resistant isolates of *L. monocytogenes* in China (7.0% (95% CI: 2.8–12.5%)) was slightly lower than that in the EU (8.1% (95% CI: 3.0–14.7%)), as shown in Table 7 (*p* < 0.01).

#### 3.3.2. Pooled Prevalence of Antibiotic Resistance of *L. monocytogenes* at Different Times

According to Table 8, the prevalence of antibiotic resistance from 2006 to 2013 was 3.2% (95% CI: 1.1–5.9%) and increased significantly to 18.9% (95% CI: 7.5–33.5%) from 2014 until 2020 (*p* < 0.01).

#### 3.3.3. Pooled Prevalence of Antibiotic Resistance of *L. monocytogenes* towards Different Antibiotics

Fifteen antibiotics were screened and meta-analyzed in this review (Table 9). The highest antibiotic resistance was found for oxacillin and ceftriaxone, at 61.2% (95% CI: 19.4–95.4%) and 27.3% (95% CI: 19.6–35.8%), respectively. The lowest antibiotic resistance was 0.0% (95% CI: 0.0–0.4%) for vancomycin, followed by ampicillin at 0.4% (95% CI: 0.0–2.9%) and erythromycin at 0.5% (95% CI: 0.0–2.3%). For a total of eight antibiotics, the pooled prevalence of *Listeria* resistance in China was lower compared to that in the EU (*p* < 0.01). On the contrary, *L. monocytogenes* resistant to ceftriaxone strains isolated from meat products in China (52.6%) were much higher than those in the EU (17.3%) (*p* < 0.01). Both China and the EU have this high rate of antibiotic resistance to oxacillin, at 61.8% and 74.8%, respectively. The resistance of *L. monocytogenes* to streptomycin and gentamicin in EU meat products was 0.0%, while resistance against these antibiotics was observed in China. However, for vancomycin, no antibiotic resistance was observed in both China and the EU. The resistance of *L. monocytogenes* towards the rest of the antibiotics between two regions were mostly at the same level.

## 4. Discussion

Food safety is a multi-faceted international issue [44]. Both developing and developed countries are actively looking for ways to prevent food safety problems. Microbial foodborne risk is usually defined as the possibility and severity of adverse effects on human health [45]. Based on the data obtained in this review, the prevalence of *L. monocytogenes* from livestock and poultry (beef, pork and chicken) meat in China remained at low levels throughout almost the last 20 years, and it is slightly lower than that in EU countries.

Chronologically, the prevalence of *L. monocytogenes* showed an overall downward trend. The prevalence peaked in 2002–2005, and gradually declined and stabilized in 2006, following the intensive regulation and legislation on food safety in China’s 10th Five-Year Plan [46] and the establishment of the General Food Law by the European Food Safety Authority in 2002 [47]. This indicates that governance at the national level plays an effective role. Among different meat products, the prevalence of *L. monocytogenes* in chicken and pork was higher than that in beef. Meanwhile, the meat consumption percentages for pork and poultry meat are above 60% and 20%, respectively, in China. Therefore, measures taken to control *L. monocytogenes* in pork and poultry meat are essential to improve the meat safety level [48].

The prevalence of *L. monocytogenes* in raw meat is, as expected, much higher than in RTE and cooked meat products. This suggests that we need to strengthen the monitoring of *L. monocytogenes* contamination in raw meat. However, the opposite is true: in recent years, more attention has been paid to foodborne pathogens in RTE and cooked meat products, while the high prevalence of *L. monocytogenes* in raw meat products has been overlooked [49,50,51]. In the vast majority of cases, raw meat is eaten after being heated or cooked; however, attention should be paid to the heating temperature, heating time, and secondary contamination. Therefore, the addition of cooking labels to different raw meat packages is a useful measure that can also effectively avoid the survival of pathogenic bacteria due to the use of incorrect processing methods or the failure to meet the required heating temperature and time. Meanwhile, RTE meat products and cooked meat products may be exposed to pathogenic bacteria and cause their spread through cross-contamination during storage, transportation, packaging, and consumption [52,53]. The food processing environment is considered to be the main source of RTE food contamination by *L. monocytogenes.* Specifically, it can survive for a long time under adverse conditions such as low temperature and is an important cause of persistent outbreaks of human listeriosis [54,55]. At the same time, the boundary between RTE meat and cooked meat is not clear in many studies, and the classification of RTE meat is not detailed. This has resulted in the inability to set appropriate subgroups for analysis. Therefore, developing a more detailed classification of RTE meat is important to determine the effect of different processed types of products with regard to prevalence and to further understand the effect of physicochemical properties on them.

The detection method of *L. monocytogenes* also has an impact on the identification results, with a slightly lower prevalence reported when biochemical identification methods were used compared to the molecular methods. The discrepancy may be due to false positives in the PCR primer design process where the amplification sequence has the same sequence as the non-target gene amplification sequence, the difficulty in distinguishing dead from live bacteria, or the false positive results due to contamination [56]. However, due to the high cost of biochemical identification and long testing period, this method is not commonly used [57]. On the contrary, molecular methods are increasingly used for the identification of *L. monocytogenes* [58]. Moreover, new techniques such as chromatography and immuno- and aptamer techniques are now being used more frequently in the detection of *L. monocytogenes*, but there is still a long way to go before they are widely used [59].

In this study, the pooled prevalence of antibiotic resistance of *L. monocytogenes* in livestock and poultry between China and EU was 5.8% (95% CI: 3.1–9.1%), which was much lower than that of RTE foods globally (38.1%; 95% CI: 36.1–39.7%) [60]. The reason for the difference may be due to the nature and diversity of the food types, their nutrient content, status, water activity, etc., as well as differences in processing methods. Internationally, RTE foods cover a wider range of food types, including fish products, dairy products, and salad products, which may undergo multiple food processing steps which increase the risk of cross-contamination. It is also relevant that under the sublethal environmental stresses encountered during the food processing, bacteria can develop a stress response and increase the resistance to the subsequent exposure to antibiotics [61,62]. In order to come to a conclusion, the number of studies regarding antibiotic resistance in the present meta-analysis, which is quite small (nine studies with 356 samples), should also be taken into account. Ampicillin and oxacillin (β-lactam antibiotics) can inhibit the synthesis of polysaccharide peptides in bacterial cell walls [63]. *L. monocytogenes* is naturally susceptible to β-lactam antibiotics, and the standard antibiotic regimen prescribed for listeriosis includes penicillin/ampicillin alone or in combination with aminoglycosides (gentamicin) [64]. The results of this study showed that *L. monocytogenes* was still sensitive to ampicillin and gentamicin, confirming the feasibility of treatment criteria. However, there was a high resistance rate (61.2%; 95% CI: 19.4–95.4%) to oxacillin. This suggests that *L. monocytogenes* has begun to develop resistance, which may be attributed to the overuse of this drug in the veterinary field [65].

In order to reduce the phenomenon of pathogenic bacteria resistance to antibiotics in animal-derived food, the EU implemented an EU-wide ban on the use of antibiotics as growth promoters in animal feed on 1 January 2006 [64]. China has also developed a corresponding catalog of banned veterinary drugs [66]. The antibiotics involved in this study, with the exception of vancomycin, are not included in the list of prohibited veterinary drugs. However, this does not imply that antibiotic resistance of pathogenic bacteria in meat products can be greatly reduced, since treatment with antibiotics may also be an option when animals are sick. During food production and processing, there is also a risk of exposure to antibiotic-resistant bacteria [67]. Taking globalization into consideration, the import and export of food between countries also contributes to the spread of antibiotic resistance [68]. The increase in antibiotic resistance rates over time was also demonstrated in the present study. To deal with this situation, governmental authorities should first strengthen the supervision of the use of antibiotics in animals bred for food production. When animals are sick, drugs other than antibiotics should be prescribed for treatment. Additionally, antibiotic susceptibility testing is necessary prior to the use of antibiotics.

## 5. Conclusions

In the present study, a review and meta-analysis was performed to compare the prevalence and antibiotic resistance of *L. monocytogenes* in livestock and poultry meat between China and the EU. Our results indicate that the overall prevalence between the two regions was slightly different, and the prevalence decreased for the last 20 years. However, the situation of antibiotic resistance of *L. monocytogenes* in livestock and poultry meat is still not optimistic, and constitutes a very serious public health issue. Currently, studies related to the prevalence of *L.* monocytogenes in meat products are dominated by qualitative sampling data, but quantitative data on its contamination levels are still lacking. Therefore, the state and enterprises should strengthen the regular investigation of *L. monocytogenes* contamination in all aspects of slaughter, processing and marketing, and establish a quantitative microbiological risk assessment of the entire chain from pasture to table to further understand the level of *L. monocytogenes* contamination.

## Figures and Tables

**Figure 1 foods-12-00769-f001:**
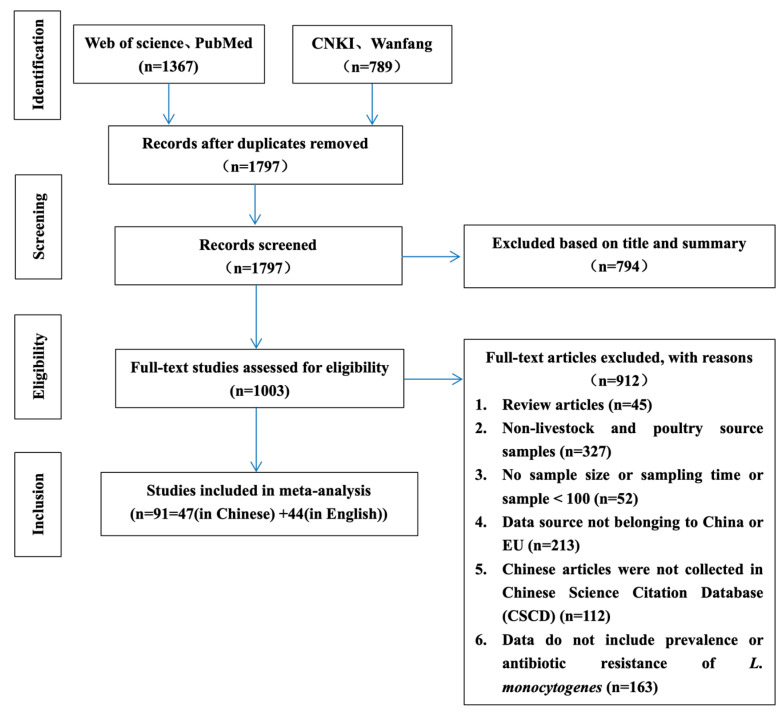
Flow diagram of the literature search and selection of eligible studies.

**Figure 2 foods-12-00769-f002:**
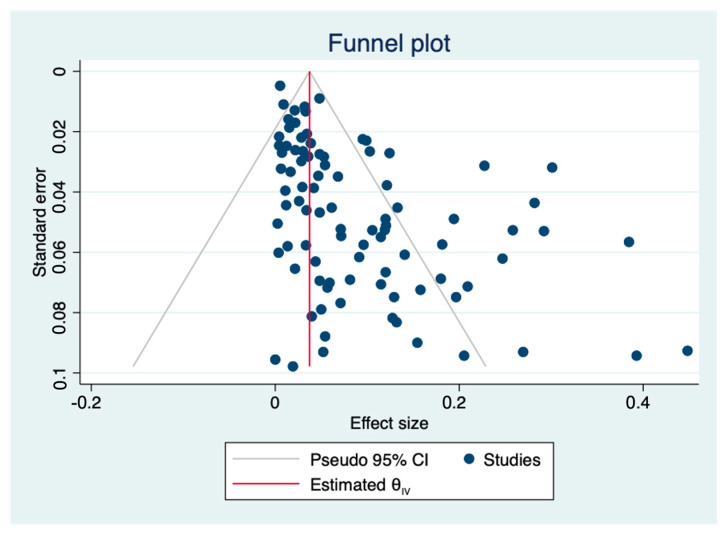
Funnel plot of the association between estimated effect sizes and standard errors in a single study of prevalence. (θ_IV_: summary estimate).

**Figure 3 foods-12-00769-f003:**
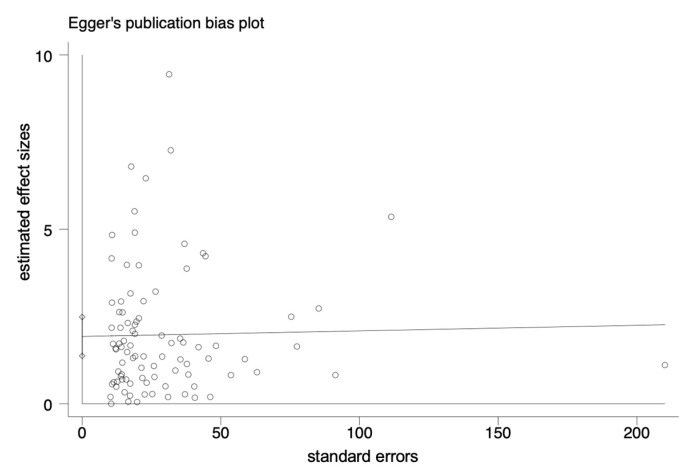
Egger plot of the association between estimated effect sizes and standard errors in a single study of prevalence.

**Figure 4 foods-12-00769-f004:**
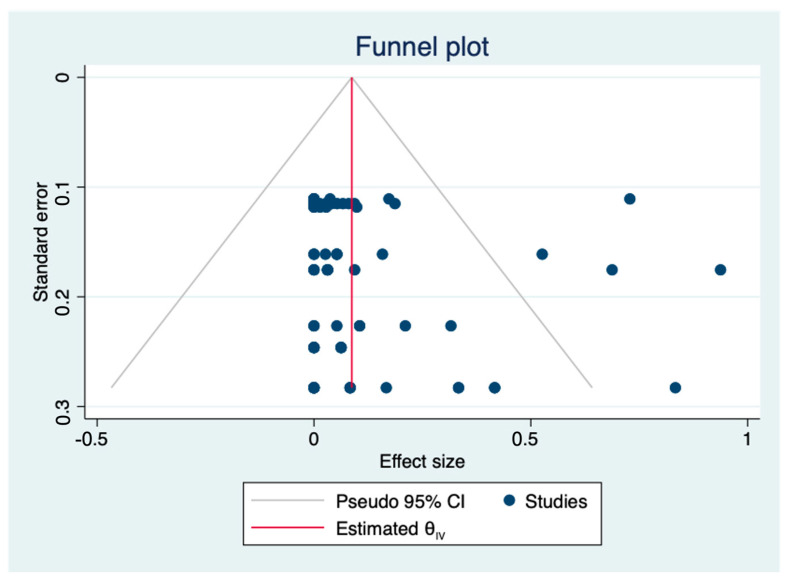
Funnel plot of the association between estimated effect sizes and standard errors in a single study of prevalence of antimicrobial resistance. (θ_IV_: summary estimate).

**Figure 5 foods-12-00769-f005:**
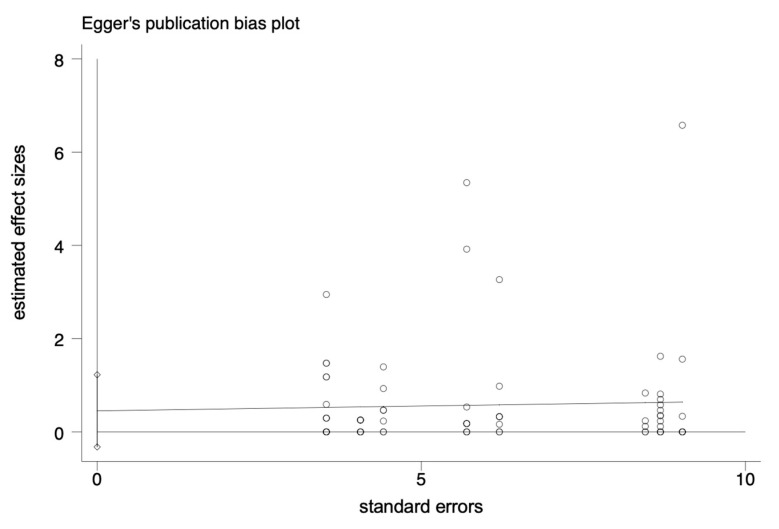
Egger plot of the association between estimated effect sizes and standard errors in a single study of the prevalence of antimicrobial resistance.

**Table 1 foods-12-00769-t001:** Search terms used in each literature database.

Literature Database	Search Terms
CNKI (www.cnki.net, in Chinese) (accessed on 18 February 2020), Wanfang Data knowledge service platform (www.wanfangdata.com.cn, in Chinese) (accessed on 18 February 2020)	((*Listeria monocytogenes*)) OR ((antibiotic-resistant) OR (prevalence) OR (contamination situation))
PubMed, Web of Science	((meat) OR (beef) OR (pork) OR (poultry)) AND (*Listeria monocytogenes*) AND ((Prevalence) OR (Prevalences) OR (Period Prevalence) OR (Period Prevalences) OR (Prevalence, Period) OR (Point Prevalence) OR (Point Prevalences) OR (Prevalence, Point)) OR ((Drug Resistances, Microbial) OR (Antimicrobial Drug Resistance) OR (Antimicrobial Drug Resistances) OR (Antibiotic Resistance, Microbial) OR (Antibiotic Resistance) OR (Resistance, Antibiotic))

**Table 2 foods-12-00769-t002:** Subgroup analysis of the prevalence of *L. monocytogenes* in different regions.

Region	No. of Studies	Pooled Prevalence	95% CI	*I*^2^ (%)	*p*-Value
China	66	7.1%	5.7–8.6%	97.69	*p* < 0.01
EU	25	8.3%	5.9–11.0%	99.27	

95% CI: 95% confidence interval. *p* < 0.01: a statistically significant prevalence in the subgroup. *I*^2^: I-squared statistics assessment of the magnitude of variation between studies.

**Table 3 foods-12-00769-t003:** Subgroup analysis of the prevalence of *L. monocytogenes* at different time periods.

Time	No. of Studies	Pooled Prevalence	95% CI	*I*^2^ (%)	*p*-Value
2000–2005	19	9.1%	6.0–12.9%	98.55	*p* < 0.01
2006–2010	34	7.4%	5.6–9.8%	98.58	
2011–2016	32	6.6%	4.6–8.8%	98.59	
2017–2020	6	6.7%	3.1–11.5%	97.72	

95% CI: 95% confidence interval. *p* < 0.01: a statistically significant prevalence in the subgroup. *I*^2^: I-squared statistics assessment of the magnitude of variation between studies.

**Table 4 foods-12-00769-t004:** Subgroup analysis of the prevalence of *L. monocytogenes* between raw meat and RTE/cooked meat.

Region	Meat Type	No. of Studies	Pooled Prevalence	95% CI	*I*^2^ (%)	*p*-Value
China	Raw meat	52	10.1%	7.8–12.7%	97.01	*p* < 0.01
	RTE/cooked meat	43	3.5%	2.6–4.5%	91.51	
EU	Raw meat	13	16.8%	9.9–24.9%	99.10	
	RTE/cooked meat	18	4.1%	2.5–6.0%	98.90	
Total	Raw meat	65	11.3%	9.0–13.8%	97.97	
	RTE/cooked meat	61	3.7%	2.9–4.7%	97.31	

95% CI: 95% confidence interval. *p* < 0.01: a statistically significant prevalence in the subgroup. *I*^2^: I-squared statistics assessment of the magnitude of variation between studies.

**Table 5 foods-12-00769-t005:** Subgroup analysis of the prevalence of *L. monocytogenes* among pork, beef and chicken.

Region	Species	No. of Studies	Pooled Prevalence	95% CI	*I*^2^ (%)	*p*-Value
China	Pork	25	11.1%	6.6–16.4%	97.22	*p* < 0.01
	Beef	19	8.7%	4.8–13.4%	86.44	
	Chicken	22	9.9%	5.8–14.8%	94.76	
EU	Pork	7	8.8%	2.1–19.2%	98.67	
	Beef	5	9.0%	3.5–16.6%	97.86	
	Chicken	4	13.9%	1.3–35.6%	96.60	
Total	Pork	32	10.5%	6.8–14.9%	97.65	
	Beef	24	8.6%	5.4–12.3%	93.01	
	Chicken	26	10.5%	6.4–15.3%	95.05	

95% CI: 95% confidence interval. *p* < 0.01: a statistically significant prevalence in the subgroup. *I*^2^: I-squared statistics assessment of the magnitude of variation between studies.

**Table 6 foods-12-00769-t006:** Subgroup analysis of the prevalence of *L. monocytogenes* by different detection methods.

Method	No. of Studies	Pooled Prevalence	95% CI	*I*^2^ (%)	*p*-Value
Biochemical identification	72	7.0%	5.7–8.4%	98.73	*p* < 0.01
Molecular detection	19	9.0%	5.7–13.0%	98.49	

95% CI: 95% confidence interval. *p* < 0.01: a statistically significant prevalence in the subgroup. *I*^2^: I-squared statistics assessment of the magnitude of variation between studies.

**Table 7 foods-12-00769-t007:** Subgroup analysis of the prevalence of antimicrobial resistance of *L. monocytogenes* in different regions.

Region	No. of Studies	Pooled Prevalence	95% CI	*I*^2^ (%)	*p*-Value
China	4	7.0%	2.8–12.5%	92.72	*p* < 0.01
EU	5	8.1%	3.0–14.7%	92.02	

95% CI: 95% confidence interval. *p* < 0.01: a statistically significant prevalence in the subgroup. *I*^2^*:* I-squared statistics assessment of the magnitude of variation between studies.

**Table 8 foods-12-00769-t008:** Subgroup analysis of the prevalence of antimicrobial resistance of *L. monocytogenes* at different time periods.

Time	No. of Studies	Pooled Prevalence	95% CI	*I*^2^ (%)	*p*-Value
2006–2013	6	3.2%	1.1–5.9%	86.77	*p* < 0.01
2014–2020	3	18.9%	7.5–33.5%	94.92	

95% CI: 95% confidence interval. *p* < 0.01: a statistically significant prevalence in the subgroup. *I*^2^: I-squared statistics assessment of the magnitude of variation between studies.

**Table 9 foods-12-00769-t009:** Subgroup analysis of the prevalence antimicrobial resistance of *L. monocytogenes* towards different antibiotics.

Region	Antibiotics	No. of Studies	Pooled Prevalence	95% CI	*I*^2^ (%)	*p*-Value
China	Oxacillin	2	61.8%	47.2–75.4%	N/A	N/A
	Ceftriaxone	1	52.6%	35.8–69.0%	N/A	N/A
	Ciprofloxacin	4	7.9%	0.0–45.0%	96.45	*p* < 0.01
	Cefotaxime	2	7.0%	2.7–12.7%	N/A	N/A
	Tetracycline	4	11.3%	4.6–20.1%	49.48	*p* > 0.05
	Doxycycline	1	5.3%	1.5–13.1%	N/A	N/A
	Chloramphenicol	4	2.7%	0.4–6.3%	0.00	*p* > 0.05
	Ampicillin	4	0.3%	0.0–3.2%	26.95	*p* > 0.05
	Erythromycin	3	1.7%	0.0–5.5%	N/A	N/A
	Streptomycin	3	3.4%	0.0–11.3%	N/A	N/A
	Cephalothin	1	4.0%	0.8–11.2%	N/A	N/A
	Rifampicin	1	4.0%	0.8–11.2%	N/A	N/A
	Trimethoprim/Sulfamethoxazole	3	3.1%	0.0–15.0%	N/A	N/A
	Vancomycin	3	0.0%	0.0–1.2%	N/A	N/A
	Gentamicin	4	2.9%	0.2–7.4%	23.19	*p* > 0.05
EU	Oxacillin	2	74.8%	65.1–83.6%	N/A	N/A
	Ceftriaxone	1	17.3%	9.8–27.3%	N/A	N/A
	Ciprofloxacin	4	9.0%	0.0–26.3%	75.64	*p* < 0.01
	Cefotaxime	1	0.0%	0.0–17.6%	N/A	N/A
	Tetracycline	4	13.0%	0.0–43.6%	89.85	*p* < 0.01
	Doxycycline	1	9.9%	4.1–19.3%	N/A	N/A
	Chloramphenicol	3	4.4%	0.0–20.1%	N/A	N/A
	Ampicillin	3	1.0%	0.0–11.0	N/A	N/A
	Erythromycin	4	0.1%	0.0–1.8%	0.00	*p* > 0.05
	Streptomycin	1	0.0%	0.0–17.6%	N/A	N/A
	Cephalothin	1	8.3%	0.2–38.5%	N/A	N/A
	Rifampicin	2	0.4%	0.0–3.9%	N/A	N/A
	Trimethoprim/Sulfamethoxazole	4	8.5%	0.0–30.8%	84.61	*p* < 0.01
	Vancomycin	4	0.0%	0.0–1.5%	23.75	*p* > 0.05
	Gentamicin	3	0.0%	0.0–1.0%	N/A	N/A
Total	Oxacillin	4	61.2%	19.4–95.4%	95.22	*p* < 0.01
	Ceftriaxone	2	27.3%	19.6–35.8%	N/A	N/A
	Ciprofloxacin	8	8.6%	0.0–26.7%	92.77	*p* < 0.01
	Cefotaxime	3	5.2%	1.6–10.2%	N/A	N/A
	Tetracycline	8	11.2%	2.7–23.3%	83.46	*p* < 0.01
	Doxycycline	2	7.4%	3.5–12.3%	N/A	N/A
	Chloramphenicol	7	2.9%	0.5–6.6%	29.97	*p* > 0.05
	Ampicillin	7	0.4%	0.0–2.9%	39.82	*p* > 0.05
	Erythromycin	7	0.5%	0.0–2.3%	0.00	*p* > 0.05
	Streptomycin	4	2.5%	0.0–8.5%	56.45	*p* > 0.05
	Cephalothin	2	3.5%	0.2–9.2%	N/A	N/A
	Rifampicin	3	6.8%	0.0–26.6%	N/A	N/A
	Trimethoprim/Sulfamethoxazole	7	5.2%	0.1–14.8%	78.47	*p* < 0.01
	Vancomycin	7	0.0%	0.0–0.4%	0.00	*p* > 0.05
	Gentamicin	7	0.9%	0.0–3.9%	32.39	*p* > 0.05

N/A: Not applicable. 95% CI: 95% confidence interval. *p* < 0.01: a statistically significant prevalence in the subgroup. *I*^2^: I-squared statistics assessment of the magnitude of variation between studies.

## Data Availability

All data related to the research are presented in the article.

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
