# Peer review of "The Prevalence and Antibiotic-Resistant of Listeria monocytogenes in Livestock and Poultry Meat in China and the EU from 2001 to 2022: A Systematic Review and Meta-Analysis"

_foods, 2023, doi:10.3390/foods12040769_

Round 1

Reviewer 1 Report

Zhang et al. conducted a meta-analysis to investigate the prevalence and antibiotic resistance rate of Listeria monocytogenes strains collected from livestock and poultry meat in China and the European Union. The analysis described in this study provided intriguing information for various aspects of the distribution and antibiotic resistance of L. monocytogenes, which could be of great interest for multiple shareholders involved in food safety, including the public health sector, food industry and scientific community. However, the current manuscript has also revealed a large room for improvement in many aspects, which is shown below.

1. Too much emphasize was placed to food safety policies in China and the EU in the Discussion section, which does not seem to be appropriate in a scientific paper. In particular, lines 290-303 and 327-336 do not help interpret the data obtained in this study, at least in the way portrayed in the current manuscript. I suggest eliminating or re-structuring them.

2. Statistical significance was not mentioned for many comparisons described in the Results section, for instance, those in lines 201-205.

3. Several statements in the Results do not match figures or tables. Please explain this discrepancy.

               Line 188: Although an increase in prevalence was observed "in the middle", it was between the last two periods (2011-2016 vs 2017-2020). Clarify this and, if necessary, revise the text.

               Lines 235 and 236: I do not think that dots are symmetrical in Figure 4. Clarify this. In Figure 4, why are there more than nine dots although the authors said that they only included nine papers in the analysis (lines 232 and 233).

4. Lines 256-258: It would be informative to readers if the authors analyzed whether this trend was observed in each region.

5. Further details are needed in the Materials and Methods section.

               Lines 122-125: How were these data extracted? By manually inspecting each paper or programmatically? Describe the process in the text.

               Lines 133 and 134: Add details for the transformation process. Were different methods tried and compared during transformation? In the meantime, the authors said in lines 136 and 137 that "Freeman-Tukey transformation was used in this meta-analysis". Clarify this and, if necessary, incorporate this information into lines 133-135.

6. References are often missing and should be provided. Some but not all examples are shown in specific comments.

7. Many statements need to be revised to improve clarity. See specific comments.

Specific comments:

Abstract

               Line 21: Change "antibiotic resistance, based on 15 antibiotics " to "resistance to 15 antibiotics".

               Line 24: Change "vs 0" to "vs 0.0%".

               Lines 24 and 25: Change "enforce good measures controlling" to "enforce good control measures against".

Introduction

               Line 31: Revise "a wide growth domain" for clarity.

               Line 32: Change "survive at" to "survive in".

               Lines 33 and 58: Add a comma before "etc.".

               Line 40: Add a comma after "Therefore" and "meat products".

               Lines 40 and 41: Provide references.

               Lines 42-44: Provide references.

               Line 50: Change "different types of meat, different time periods" to "different types of meat and time periods".

               Lines 48-51: Provide references.

               Line 55: Remove the comma before "(OIE)". Change "collect and provide" to "have collected and provided".

               Lines 56 and 57: "warn about the control of them" does not make sense; please revise it.

               Lines 66 and 67: Revise "also lacks data supporting" to improve clarity.

               Line 68: Change "to the field of" to "in the field of".

               Lines 69 and 70: Change "prevalence, incidence" to "prevalence".

               Line 72: I recommend removing "in a certain research area".

               Line 77: Change "raw meat and RTE meat products" to "raw and RTE meat products".

               Line 80: Change "are provided" to "was provided" and revise "only certain .. up to now", which sounds vague.

               Lines 81-83: Re-write "China and the European Union ... producers" to improve clarity.

               Line 89: Change "Based on quantitative analysis" to "In the quantitative analysis".

               Line 90: Show the acronym "EU" when "European Union" was first shown in line 81. Check the wording for "the source of the differences can be identified".

Materials and Methods

               Lines 97 and 98: Move "Preferred ... 2022" to the References section.

               Line 100: Change "for literature retrieval were used" and "PubMed and Web of Science" to "were used for literature retrieval" and "PubMed, Web of Science", respectively.

               Lines 101 and 102: Remove the underscore in "in Chinese".

               Lines 103 and 113: I recommend creating a table that shows the keywords used to retrieve papers.

               Lines 106 and 107: What does "with close ... and close synonyms" mean? Clarify and re-write it.

               Line 115: I recommend changing "established" to "applied".

               Lines 116-119: Change "was clearly stated" to "should be clearly stated". Revise other similar instances as well.

               Line 118: Change "antibiotic-resistant of" to "antibiotic resistance of".

               Line 119: This is confusing since CSCD was not mentioned in lines 100-102.

               Line 123: Change "Author name" to "author name".

               Line 124: Clarify and revise "L. monocytogenes positive strains". Add "and" before "antibiotic resistance".

               Lines 125 and 126: Clarify "the opinions were ... validation". In fact, is this really necessary?

               Line 126: Change "Th extracted data was" to "The extract data were".

               Line 130: Change "extracted from meat" to "isolated from meat".

               Line 132: Add a comma before "were meta-analyzed".

               Line 136: Show the manufacturer of Stata17 and its geographical location.

               Line 145: Change "the Metan and the Metaprop is usually used" to "the Metan and Metaprop were usually used".

               Lines 147 and 148: Change "cannot be used" and "will be excluded" to "could not be used" and "are excluded", respectively.

Results

               Lines 154 and 155: Please include this process in the Materials and Methods section.

               Lines 157 and 158: Condense "47 Chinese literatures and 44 English literatures".

               Lines 166 and 235: Explain what "high heterogeneity" means and revise the text accordingly.

               Line 168: Remove the comma after "Fig. 3".

               Line 169: Revise "P < 0.05".

               Line 176: Italicize "L. monocytogenes". Revise similar instances in the headings throughout the manuscript.

               Line 184: Change "was -changing" to "was changing".

               Line 185: Change the wording for "subgroups were established within four different time periods".

               Line 216: Change "true for chicken" to "observed for chicken".

               Line 224: I recommend changing "identified" to "regarded".

               Lines 265-268: Table 8 is not appropriate for this statement and I recommend removing it.

               Line 273: Revise "less resistant in China compared to the EU" for clarity since this could give an impression that this pertains to magnitude. Also, is this statistically significant?

               Lines 273-277: Revise these statements so that it can be clear that they pertain to frequency.

               Lines 275 and 276: Change "streptomycin, gentamicin" to "streptomycin and gentamicin".

Discussion

               Line 285: Is "multi-faced" a typo for "multi-faceted"? Provide references.

               Line 293: Change "laws and regulations enforcement" to "law and regulation enforcement".

               Lines 300 and 301: Provide references.

               Line 304: Change "In terms of time" to "Chronologically".

               Lines 310-312: Revise these statements for clarity. What do percentages represent? A focus for what?

               Lines 314 and 315: Clarify "focus on L. monocytogenes in raw meat".

               Line 315: Replace the comma after "true" with a colon.

               Line 316: Change "of raw" to "of L. monocytogenes in raw".

               Line 318: Replace the comma after "cooked" with a semi-colon.

               Lines 320 and 321: Revise "effectively avoid microbial residues due to insufficient processing" for clarity.

               Lines 322 and 323: Revise "cause microbial proliferation" for clarity.

               Lines 324-327: Start a new sentence from "especially". Replace "persistence" and "persistent" with other terms since persistence often means inhabitation of a certain clone within a processing facility for an extended period of time.

               Line 330: Does "food" mean RTE foods?

               Lines 330-332: Provide references.

               Line 332: Revise "RTE foods that promote their growth" for clarity.

               Lines 333 and 334: Clarify and re-write "the product does not ... shelf life".

               Lines 345 and 346: Clarify and re-write "the homology ... primer design".

               Lines 347 and 348: Provide references.

               Lines 353, 365 and 370-371: Italicize "L. monocytogenes".

               Lines 355 and 356: Further articulate this statement with further details.

               Lines 356 and 357: Change "The Internationally" to "Internationally".

               Lines 359-360: Start a separate sentence from "given also" in line 359. In fact, I have doubts whether this part ("given also ... antibiotics") is relevant in discussing the findings obtained from this study.

               Line 362: Change "come to conclusion" to "come to a conclusion".

               Line 363: Add commas before and after "which is quite small (9 studies with 356 samples)".

               Line 364: Is "-lactam antibiotics" a typo for "beta-lactam antibiotics"?

               Line 367: Explain why glucosamine and Gentamicin are shown together. Change "Gentamicin" to "gentamicin".

               Lines 368-370: Change "Ampicillin and Gentamicin" and "Oxacillin" to "ampicillin and gentamicin" and "oxacillin", respectively.

               Lines 371 and 372: I believe that additional supporting data should be provided to make this statement ("is attributed to ... field").

               Lines 373-374 and 378: Re-write "animal-derived ... antibiotics" and "antibiotic resistance in meat products", which do not make sense given that bacteria, not foods, are resistant to antibiotics.

               Lines 375 and 376: Provide references.

               Line 387: Change "prior" to "prior to".

Conclusions

               Line 391: Change "There results" to "Our results".

               Line 392: Change "the last 20 years" to "for the last 20 years".

               Line 394: "(the situation) has increased greatly in the past 20 years"? Re-write this part for clarity.

               Lines 397-399: This statement has little to with the prior statements and does not fit into the context.

               Lines 399-402: This statement is incomplete. Revise it.

Figure 1

               Correct spelling errors in the figure, including "aticles", "atticles", and "sourse". Change "Exclude based" to "Excluded based".

Figure 2 and other similar figures

               Explain what estimated theta IV represents.       

Figure 3 and other similar figures

               Line 174: Do "estimated effect sizes and standard errors" correspond to y and x axes, respectively? I recommend keeping axis labels and the legend consistent.

Table 1 and other similar tables

               Please show what P-value represents here. If P-value is less than 0.05, does this mean that prevalences shown in the table or a subgroup are not identical?

Table 5

               Change "Molecular" to "Molecular detection".

Table 8

               Change "0" in "Pooled prevalence" column to "0.0%".    

Reviewer 2 Report

The title seems adequate, but it must specify the meat species that are included in the review (beef, pork, and chicken).

The design of the methodology seems convenient for this type of study and the inclusion/exclusion criteria, the selection of the information in the realization of the database for the meta-analysis and the statistical analysis are perfectly established. The discussion and conclusion are clear.

Line 42: specify the abbreviation RTE (Ready to Eat).

Line 96-102: Change the format of the citations, for example: (Cumpston et al., 2019) by (35) and successively the URL citations too.

Line 160: Check the spelling of the information in the boxes of the literature selection flow diagram (Identificatio, aticles, sourse, atttiticles).

Reviewer 3 Report

The article written by Zhang et al. describes meta-analysis of prevalence and antibiotic resistance rate of Listeria monocytogenes in livestock and poultry meat. Furthermore, the authors compare the results between China and European Union countries.

In my opinion the article is well written. Analyses were conducted based on properly selected statistical methods to calculate an overall effect.
Furthermore, the whole construction of the article is also well. The discussion part makes a fine complement to the whole story.

I have only one minor issue for the authors: in line 55 it was used the obsolete organisation name- Office Ιnternational des Εpizooties, (OIE), please change it for World Organisation for Animal Health (WOAH).
